# Consistency Regularization for Adversarial Robustness

**Jihoon Tack**[1]  **Sihyun Yu**[1]  **Jongheon Jeong**[1]  **Minseon Kim**[1]  **Sung Ju Hwang**[1 2]  **Jinwoo Shin**[1]

## Abstract

*Adversarial training* (AT) is currently one of the most successful methods to obtain the adversarial robustness of deep neural networks. However, the phenomenon of robust overfitting, *i.e.*, the robustness starts to decrease significantly during AT, has been problematic, not only making practitioners consider a bag of tricks for a successful training, *e.g.*, early stopping, but also incurring a significant generalization gap in the robustness. In this paper, we propose an effective regularization technique that prevents robust overfitting by optimizing an auxiliary 'consistency' regularization loss during AT. Specifically, it forces the predictive distributions after attacking from two different augmentations of the same instance to be similar with each other. Our experimental results demonstrate that such a simple regularization technique brings significant improvements in the test robust accuracy of a wide range of AT methods. More remarkably, we also show that our method could significantly help the model to generalize its robustness against unseen adversaries, *e.g.*, other types or larger perturbations compared to those used during training.

## 1. Introduction

Recent studies have demonstrated that deep neural networks (DNNs) are vulnerable to adversarial examples, *i.e.*, inputs crafted by an imperceptible perturbation which confuse the network prediction (Szegedy et al., 2014; Goodfellow et al., 2015). This vulnerability of DNNs raises serious concerns about their deployment in the real-world (Chen et al., 2015; Kurakin et al., 2016; Li et al., 2019).

*Adversarial training* (AT) is currently one of the most promising ways to obtain the adversarial robustness of

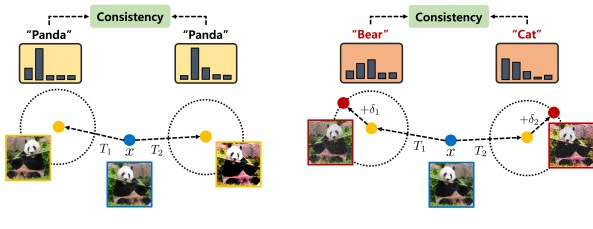

(a) Previous CR  (b) Proposed CR

*Figure 1.* An overview of the previous and proposed consistency regularization (CR) scheme. We force the predictive distribution of *attacked* augmentations to be consistent. $T$ and $\delta$ denotes the randomly sampled augmentation, and the corresponding adversarial noise, respectively.

DNNs, *i.e.*, directly augmenting the training set with adversarial examples (Goodfellow et al., 2015; Madry et al., 2018; Athalye et al., 2018). However, one of the major downsides that most AT methods suffer from, is a significant generalization gap of adversarial robustness between the train and test datasets (Yang et al., 2020). More importantly, it has been observed that such a gap gradually increases from the middle of training (Rice et al., 2020), *i.e.*, robust overfitting, which makes practitioners to consider heuristic approaches for a successful optimization, *e.g.*, early stopping (Zhang et al., 2019). Only recently, a few proposed more advanced regularization techniques, *e.g.*, self-training (Chen et al., 2021) and weight perturbation (Wu et al., 2020), but it is still largely unknown to the community that why and how only such sophisticated training schemes could be effective to prevent the robust overfitting of AT.[1]

**Contribution.** In this paper, we suggest to optimize an auxiliary '*consistency*' regularization loss, as a simpler and easy-to-use alternative for regularizing AT. To this end, we first found that the existing data augmentation (DA) schemes are already quite effective to reduce the robust overfitting in AT. Upon the observation, we optimize a consistency regularization loss during AT which forces *adversarial examples* from two independent augmentations of the same input to have similar predictions. Here, it looks highly non-trivial at first glance whether matching such attack directions over DA is useful in any sense. Our finding is that the attack direction provides intrinsic information of the sample (other than

---

[1]Korea Advanced Institute of Science and Technology (KAIST), Daejeon, South Korea [2]AITRICS, Seoul, South Korea. Correspondence to: Jinwoo Shin <jinwoos@kaist.ac.kr>.

*Accepted by the ICML 2021 workshop on A Blessing in Disguise: The Prospects and Perils of Adversarial Machine Learning.* Copyright 2021 by the author(s).

---

[1]A concurrent work with ours, hypothesize the robust overfitting issue as a DNN's memorization issue (Dong et al., 2021).

its label), where the most frequently attacked class is the most confusing class of the 'clean' input, *i.e.*, class with the maximum softmax probability disregarding the true label. The proposed regularization loss injects a strong inductive bias to the model that such 'dark' knowledge (Hinton et al., 2015) over DA should be consistent.

We verify the efficacy of our scheme through extensive evaluations on CIFAR-10/100 (Krizhevsky et al., 2009) and Tiny-ImageNet. For example, our regularization could improve the robust accuracy of WideResNet (Zagoruyko and Komodakis, 2016) trained via standard AT (Madry et al., 2018) on CIFAR-10 from 45.62%→50.37%. Moreover, we show that our regularization could even notably improve the robustness against unforeseen adversaries (Tramer and Boneh, 2019), *i.e.*, when the adversaries assume different threat models from those used in training: *e.g.*, our method could improve the $l_1$-robustness of TRADES (Zhang et al., 2019) from 28.64%→46.73% on PreAct-ResNet (He et al., 2016). Finally, we also observe that our method could be even beneficial for the corruption robustness (Hendrycks and Dietterich, 2019).

## 2. Consistency regularization for adversarial robustness

### 2.1. Preliminaries: Adversarial training

We consider a classification task with a given $K$-class dataset $\mathcal{D} = \{(x_i, y_i)\}_{i=1}^n \subseteq \mathcal{X} \times \mathcal{Y}$, where $x \in \mathbb{R}^d$ represents an input sampled from a certain data-generating distribution $P$ in an *i.i.d.* manner, and $\mathcal{Y} := \{1, \ldots, K\}$ represents a set of possible class labels. Let $f_\theta : \mathbb{R}^d \to \Delta^{K-1}$ be a neural network modeled to output a probability simplex $\Delta^{K-1} \in \mathbb{R}^K$, *e.g.*, via a softmax layer. Concretely, the adversarial robustness we primarily focus in this paper is the $\ell_p$-*robustness*: *i.e.*, for a given $p \geq 1$ and a small $\epsilon > 0$, we aim to train a classifier $f_\theta$ that correctly classifies $(x + \delta, y)$ for any $\|\delta\|_p \leq \epsilon$, where $(x, y) \sim P$.

The high level idea of *adversarial training* (AT) is to directly incorporate adversarial examples to train the classifier (Goodfellow et al., 2015). In general, AT methods formalize the training of $f_\theta$ as an alternative min-max optimization with respect to $\theta$ and $\|\delta\|_p \leq \epsilon$, respectively; *i.e.*, one minimizes a certain the classification loss $\mathcal{L}$ with respect to $\theta$ while an adversary maximizes $\mathcal{L}$ by perturbing the given input to $x + \delta$ during training. Here, for a given $\mathcal{L}$, we denote the inner maximization procedure of AT as $\mathcal{L}_{\mathrm{adv}}(x, y; \theta)$:

$$\mathcal{L}_{\mathrm{adv}}(x, y; \theta) := \max_{\|\delta\|_p \leq \epsilon} \mathcal{L}(x, y, \delta; \theta). \tag{1}$$

### 2.2. Effect of augmentations in adversarial training

Now, we investigate the utility of data augmentations in AT. Throughout this section, we train PreAct-ResNet-18 (He

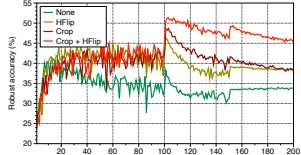 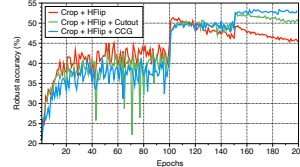

(a) Conventional augmentations     (b) Additional augmentations

*Figure 2.* Robust accuracy (%) against PGD-10 attack on standard AT (Madry et al., 2018) under (a) conventional augmentations, and (b) additional augmentations to the convention. We consider PreAct-ResNet-18 trained on CIFAR-10. We use $l_\infty$ threat model with $\epsilon = 8/255$. None, HFlip, and Crop, indicates no augmentation, horizontal flip, and random crop, respectively. CCG denotes the combined augmentation of Cutout, color jitter, and gray scale.

et al., 2016) on CIFAR-10 (Krizhevsky et al., 2009) using standard AT (Madry et al., 2018), following the training details of Rice et al. (2020). We use projected gradient descent (PGD) with 10 iterations under $\epsilon = 8/255$ (step size of $2/255$) with $l_\infty$ constraint to perform adversarial attacks for both training and evaluation. Formally, for a given training sample $(x, y) \sim \mathcal{D}$, and augmentation $T \sim \mathcal{T}$, the training loss is:

$$\max_{\|\delta\|_\infty \leq \epsilon} \mathcal{L}_{\mathrm{CE}}(f_\theta(T(x) + \delta), y). \tag{2}$$

Unless otherwise specified, we assume the set of baseline augmentations $\mathcal{T} := \mathcal{T}_{\mathrm{base}}$ (*i.e.*, random crop with 4 pixels zero padding and horizontal flip) by default for this section.

**Role of base augmentations in adversarial training.** We observe that the base augmentations $\mathcal{T}_{\mathrm{base}}$ are already somewhat useful for relaxing the robust overfitting in AT. To this end, we conduct a controlled experiment by removing each augmentation from the pre-defined augmentation set $\mathcal{T}_{\mathrm{base}}$ and train the network. Figure 2a summarizes the result of the experiment. As each augmentation is removed, not only the robustness degrades, but also the adversarial overfitting is getting significant. This result implies that there may exist an augmentation family that effectively prevents the robust overfitting as the base augmentation is already useful.

**Reducing robust overfitting with data augmentations.** We further find that the existing data augmentation schemes are already quite effective to reduce the robust overfitting in AT. Specifically, we additionally utilize Cutout (DeVries and Taylor, 2017), color distortion, and gray scale augmentation, which are commonly used in other vision domains (Chen et al., 2020; Khosla et al., 2020) (we denote such augmentation family as $\mathcal{T}_{\mathrm{CCG}}$). As shown in Figure 2b, the robust overfitting is gradually reduced as more diverse augmentations are used, and even the best accuracy improves.

### 2.3. Consistency regularization for adversarial training

We suggest to optimize a simple auxiliary *consistency regu-*

*Table 1.* Comparison of the consistency regularization (CR) loss. We report clean accuracy and robust accuracy (%) against PGD-100 attack of PreAct-ResNet-18 trained on CIFAR-10. We use $l_\infty$ threat model with $\epsilon = 8/255$.

| Loss | Clean | PGD-100 |
|---|---|---|
| AT (2) | 83.90 | 53.22 |
| AT (2) + previous CR (4) | 84.91 | 53.14 |
| AT (2) + proposed CR (3) | 84.65 | 54.67 |

*larization* during AT that further improve the robust generalization. Specifically, our regularization forces *adversarial examples* from two independent augmentations of an instance to have a similar prediction (see Figure 1). It is highly non-trivial whether matching such attack directions via consistency regularization is useful, which we essentially investigate in this paper. Our major finding is that the attack direction itself contains intrinsic information of the instance, as in Appendix E. For example, the most frequently attacked class is the most confusing class of the 'clean' input, *i.e.*, class with the maximum softmax probability disregarding the true label. Hence, our regularization utilize this dark knowledge (other than the true labels) of samples and induce a strong inductive bias to the classifier.

Formally, for a given data point $(x, y) \sim \mathcal{D}$ and augmentations $T_1, T_2 \sim \mathcal{T}$, we denote $\delta_i$ as an adversarial noise of $T_i(x)$, *i.e.*, $\delta_i := \arg\max_{\|\delta\|_p \le \epsilon} \mathcal{L}(T_i(x), y, \delta; \theta)$. We consider regularizing the temperature-scaled distribution $\hat{f}_\theta(x; \tau)$ (Guo et al., 2017) over the adversarial examples across augmentations to be consistent, where $\tau$ is the temperature hyperparameter; $\hat{f}_\theta(x; \tau) = \text{Softmax}(z_\theta(x)/\tau)$ where $z_\theta(x)$ is the logit value of $f_\theta(x)$, *i.e.*, activation before the softmax layer of $f_\theta(x)$. Then the proposed regularization loss is given by:

$$\text{JS}\Big(\hat{f}_\theta\big(T_1(x) + \delta_1; \tau\big) \parallel \hat{f}_\theta\big(T_2(x) + \delta_2; \tau\big)\Big), \quad (3)$$

where $\text{JS}(\cdot \parallel \cdot)$ denotes the Jensen-Shannon divergence. Since the augmentations are randomly sampled in every training step, adversarial example's predictions become consistent regardless of augmentation selection when minimizing the proposed objective. We note that the motivation behind the temperature scaling is that the confidence of prediction (*i.e.*, maximum softmax value) is relatively low on AT than the standard training. Hence, we compensate this issue by enforcing the sharp distribution by using a small temperature.

**Comparison to other consistency regularization loss over DA.** There has been prior works that suggested a consistency regularization loss to better utilize DA (Hendrycks et al., 2020; Zhang et al., 2020; Sohn et al., 2020), which can be expressed with the following form:

$$\text{D}\Big(f_\theta\big(T_1(x)\big), f_\theta\big(T_2(x)\big)\Big), \quad (4)$$

where D is a discrepancy function. The regularization term used in (4) has a seemingly similar formula to ours but there is a fundamental difference: our method (3) does not match the predictions directly for the 'clean' augmented samples, but does after *attacking* them independently, *i.e.*, $f_\theta(T(x) + \delta)$. To examine which one is better, we compare (3) with (4) under the same discrepancy function, $\text{D} := \text{JS}$ and same augmentation familiy $\mathcal{T}_{\text{CCG}}$. As shown in Table 1, our design choice (3) improves both clean and robust accuracy compare to the baseline (2), while the prior consistency regularization (4) only improves the clean accuracy.

**Overall training objective.** In the end, we derive a final training objective, $\mathcal{L}_{\text{total}}$: an AT objective combined with the consistency regularization loss (3). To do so, we consider the average of inner maximization objective on AT $\mathcal{L}_{\text{adv}}$ (1) over two independent augmentations $T_1, T_2 \sim \mathcal{T}$, as minimizing (1) over the augmentations $T \sim \mathcal{T}$ is equivalent to an average of (1) over $T_1$ and $T_2$:

$$\frac{1}{2}\Big(\mathcal{L}_{\text{adv}}\big(T_1(x), y; \theta\big) + \mathcal{L}_{\text{adv}}\big(T_2(x), y; \theta\big)\Big). \quad (5)$$

We then combine our regularizer (3) with a given hyperparameter $\lambda$, into the average of inner maximization objectives (5). Then the final training objective $\mathcal{L}_{\text{total}}$ is as follows:

$$\begin{aligned}\mathcal{L}_{\text{total}} := &\frac{1}{2}\Big(\mathcal{L}_{\text{adv}}\big(T_1(x), y; \theta\big) + \mathcal{L}_{\text{adv}}\big(T_2(x), y; \theta\big)\Big) \\ &+ \lambda \cdot \text{JS}\Big(\hat{f}_\theta\big(T_1(x) + \delta_1; \tau\big) \parallel \hat{f}_\theta\big(T_2(x) + \delta_2; \tau\big)\Big)\end{aligned}$$

Note that our regularization scheme is agnostic to the choice of AT objective, hence, can be easily incorporated into well-known AT methods (Madry et al., 2018; Zhang et al., 2019; Wang et al., 2020). We introduce explicit forms of other variants of final objective $\mathcal{L}_{\text{total}}$ for various AT methods, integrated with our regularization loss, in Appendix A.

## 3. Experiments

We verify the effectiveness of our simple technique on image classification datasets: CIFAR-10/100 (Krizhevsky et al., 2009) and Tiny-ImageNet. Our results exhibit that incorporating simple consistency regularization scheme into the existing adversarial training (AT) methods significantly improve adversarial robustness against various attack methods (Carlini and Wagner, 2017; Madry et al., 2018; Croce and Hein, 2020b), including data corruption (Hendrycks and Dietterich, 2019). Intriguingly, our method shows better robustness against *unseen* adversaries compared to other baselines. Moreover, our method shows comparable or somewhat surpass the performance of the recent regularization technique (Wu et al., 2020) (see Appendix D). The detailed experimental setups, *e.g.*, training details, datasets, and hyperparameters, are specified in Appendix C.

*Table 2.* Clean accuracy and robust accuracy (%) against diverse attacks of each individual, and combined regularization. The numbers below the attack methods, indicate the radius of the perturbation $\epsilon$. All results are reported on PreAct-ResNet-18 trained under various image classification benchmark datasets. The bold indicates the improved results by our regularization loss.

| Dataset | Method | Clean | $l_\infty$ (Seen) | | | $l_2$ (Unseen) | | $l_1$ (Unseen) | |
|---|---|---|---|---|---|---|---|---|---|
| | | | PGD-100 (8/255) | CW$_\infty$ (8/255) | AutoAttack (8/255) | PGD-100 (150/255) | PGD-100 (300/255) | PGD-100 (2000/255) | PGD-100 (4000/255) |
| CIFAR-10 | Standard (Madry et al., 2018) | 84.48 | 45.89 | 45.08 | 40.74 | 52.67 | 26.91 | 43.44 | 32.44 |
| | **+ Consistency** | **84.65** | **54.67** | **51.32** | **47.83** | **62.60** | **34.43** | **54.52** | **42.45** |
| | TRADES (Zhang et al., 2019) | 82.20 | 51.13 | 49.04 | 46.41 | 55.91 | 28.31 | 42.36 | 28.64 |
| | **+ Consistency** | **83.18** | **54.68** | **50.50** | **48.30** | **61.46** | **37.11** | **56.09** | **46.73** |
| | MART (Wang et al., 2020) | 80.41 | 49.41 | 45.59 | 41.89 | 55.80 | 30.15 | 43.58 | 27.00 |
| | **+ Consistency** | 80.09 | **55.16** | **50.17** | **47.02** | **60.65** | **38.10** | **54.85** | **43.29** |
| CIFAR-100 | Standard (Madry et al., 2018) | 56.96 | 20.86 | 21.20 | 18.93 | 27.65 | 11.08 | 26.49 | 21.48 |
| | **+ Consistency** | **60.21** | **28.27** | **26.44** | **23.71** | **36.17** | **16.77** | **37.00** | **33.56** |
| Tiny-ImageNet | Standard (Madry et al., 2018) | 41.10 | 10.93 | 10.79 | 9.20 | 27.84 | 17.71 | 32.62 | 30.91 |
| | **+ Consistency** | **45.61** | **17.71** | **16.43** | **13.56** | **34.78** | **28.36** | **38.36** | **36.40** |

*Table 3.* Mean corruption error (mCE) (%) of PreAct-ResNet-18 trained on CIFAR-10, and tested with CIFAR-10-C dataset (Hendrycks and Dietterich, 2019). The bold indicates the improved results by the proposed method.

| Method | mCE ↓ |
|---|---|
| Standard (Madry et al., 2018) | 24.05 |
| **+ Consistency** | **22.06** |
| TRADES (Zhang et al., 2019) | 26.17 |
| **+ Consistency** | **24.05** |
| MART (Wang et al., 2020) | 27.76 |
| **+ Consistency** | **26.75** |

**Evaluation setup.** Throughout the section, we mainly report the results where the clean accuracy converges, *i.e.*, fully trained model, to focus on the robust overfitting problem (Rice et al., 2020). Nevertheless, we also note that our regularization method achieves better best robust accuracy compare to the AT methods (see Table 4 in Appendix D).

**Considered adversarial attack.** We consider a wide range of adversarial attacks in order to measure the robustness of models without gradient obfuscation (Athalye et al., 2018): PGD (Madry et al., 2018) with 100 iterations (step size with $2\epsilon/k$, where $k$ is the iteration number), CW$_\infty$ (Carlini and Wagner, 2017), and AutoAttack (Croce and Hein, 2020b).

### 3.1. Main results

**Seen adversaries.** As shown in Table 2, incorporating our regularization scheme into existing AT methods consistently improves accuracies against various adversaries across different models and datasets. In particular, for standard AT, our method relatively improves 17.40% of robust accuracy against the AutoAttack. More intriguingly, consideration of our regularization technique into the AT methods boosts the clean accuracy as well in most cases. We notice that such improvement is non-trivial, as some works have reported a trade-off between a clean and robust accuracies in AT (Tsipras et al., 2019; Zhang et al., 2019).

**Unseen adversaries.** We evaluate our method against *unforeseen* adversaries, *e.g.*, robustness on different norm constraints of $l_2$ and $l_1$, as reported in Table 2. We observe that combining our regularization method could consistently and significantly improve the robustness against all the considered unseen adversaries tested. It is remarkable that our method is especially effective against $l_1$ adversaries compared to the baselines, regarding the fundamental difficulty of achieving the mutual robustness against both $l_1$ and $l_\infty$ attacks (Tramer and Boneh, 2019; Croce and Hein, 2020a). We also show that our regularization scheme improves on different types of unseen adversaries: attack under different radii of $\epsilon$ with $l_\infty$ in Appendix D.

**Common corruption.** We also validate the effectiveness of our method on corrupted CIFAR-10 dataset (Hendrycks and Dietterich, 2019), *i.e.*, consist of 19 types of corruption such as snow, zoom blur. We report the mean corruption error (mCE) of each model in Table 3. The results show that the mCE consistently improves combined with our regularization loss regardless of AT methods. Interestingly, our method even reduces the error (from the standard cross-entropy training) of corruptions that are not related to the applied augmentation or noise, *e.g.*, zoom blur error 25.1%→20.2%. We note that common corruption is also important and practical defense scenario (Hendrycks and Dietterich, 2019), therefore, obtaining such robustness should be a desirable property for a robust classifier.

## 4. Conclusion

In this paper, we propose a simple yet effective regularization technique to tackle the robust overfitting in adversarial training (AT). Our regularization forces the predictive distributions after attacking from two different augmentations of the same input to be similar to each other. Our experimental results demonstrate that the proposed regularization brings significant improvement in various defense scenarios including unseen adversaries.

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

# Appendix

## Consistency Regularization for Adversarial Robustness

## A. Overview on adversarial training objectives

In this section, we provide an overview of adversarial training (AT) objectives: standard AT (Madry et al., 2018), TRADES (Zhang et al., 2019), and MART (Wang et al., 2020). We use the same notation as in Section 2.

### A.1. Standard adversarial training

One of the most basic form of AT method (Madry et al., 2018) considers to design $\mathcal{L}_{\text{adv}}$ with the standard cross-entropy loss $\mathcal{L}_{\text{CE}}$, and directly force the generated adversarial example to have the original label. Formally, for a given sample $(x, y) \sim \mathcal{D}$, a classifier $f_\theta$ and a ball constraint with $\epsilon$ the loss is as follows:

$$\mathcal{L}_{\text{AT}} := \max_{\|\delta\|_p \leq \epsilon} \mathcal{L}_{\text{CE}}\big(f_\theta(x + \delta), y\big). \tag{6}$$

**Standard AT with consistency regularization.** By considering standard AT loss as the AT objective, *i.e.*, $\mathcal{L}_{\text{adv}} = \mathcal{L}_{\text{AT}}$ (6), one can define the final objective $\mathcal{L}_{\text{total}}^{\text{AT}}$. For a given data augmentations $T_1, T_2 \sim \mathcal{T}$ and sharpening temperature $\tau$, the final objective is as follows:

$$\mathcal{L}_{\text{total}}^{\text{AT}} = \frac{1}{2} \sum_{i=1}^{2} \max_{\|\delta_i\|_p \leq \epsilon} \mathcal{L}_{\text{CE}}\big(f_\theta(T_i(x) + \delta_i), y\big) + \lambda \cdot \text{JS}\Big(\hat{f}_\theta\big(T_1(x) + \delta_1; \tau\big) \parallel \hat{f}_\theta\big(T_2(x) + \delta_2; \tau\big)\Big), \tag{7}$$

where $\delta_i = \arg\max_{\|\delta\|_p \leq \epsilon} \mathcal{L}_{\text{CE}}(f_\theta\big(T_i(x) + \delta\big), y)$, and $\hat{f}_\theta(x; \tau)$ is the temperature scaled classifier.

### A.2. TRADES

Zhang et al. (2019) showed that there can exist a trade-off between clean and adversarial accuracy and decomposed the objective of standard AT into clean and robust objectives. By combining two objectives with a balancing hyperparameter, one can control such trade-offs. For a given sample $(x, y) \sim \mathcal{D}$, and a classifier $f_\theta$ the proposed training objective is:

$$\mathcal{L}_{\text{TRADES}} := \mathcal{L}_{\text{CE}}\big(f_\theta(x), y\big) + \beta \cdot \max_{\|\delta\|_p \leq \epsilon} \text{KL}\big(f_\theta(x) \parallel f_\theta(x + \delta)\big), \tag{8}$$

where $\text{KL}(\cdot \parallel \cdot)$ denotes the Kullback–Leibler divergence, $\mathcal{L}_{\text{CE}}$ is the cross-entropy loss, and $\beta$ is the hyperparameter to control the trade-off between clean accuracy and robust accuracy. For all experiments, we set the hyperparameter $\beta = 6$ by following the original paper.

**TRADES with consistency regularization.** By utilizing $\mathcal{L}_{\text{TRADES}}$ as the base AT objective, one can also adapt our regularization scheme for the final objective $\mathcal{L}_{\text{total}}^{\text{TRADES}}$. We also apply the consistency regularization loss between adversarial examples with the Jensen-Shannon divergence, $\text{JS}(\cdot \parallel \cdot)$. For given sampled data augmentations $T_1, T_2 \sim \mathcal{T}$ and sharpening temperature $\tau$, the final objective is as follows:

$$\begin{aligned} \mathcal{L}_{\text{total}}^{\text{TRADES}} = \frac{1}{2} \sum_{i=1}^{2} \Big( \mathcal{L}_{\text{CE}}\big(f_\theta(T_i(x)), y\big) + \beta \cdot \text{KL}\big(f_\theta(T_i(x)) \parallel f_\theta(T_i(x) + \delta_i)\big)\Big) \\ + \lambda \cdot \text{JS}\Big(\hat{f}_\theta\big(T_1(x) + \delta_1; \tau\big) \parallel \hat{f}_\theta\big(T_2(x) + \delta_2; \tau\big)\Big), \end{aligned} \tag{9}$$

where $\delta_i = \arg\max_{\|\delta\|_p \leq \epsilon} \text{KL}\big(f_\theta(T_i(x)) \parallel f_\theta(T_i(x) + \delta)\big)$, and $\hat{f}_\theta(x; \tau)$ is the temperature scaled classifier (Guo et al., 2017).

### A.3. MART

Wang et al. (2020) observed that addressing more loss on the misclassified sample during training can improve the robustness. Let $f_\theta^{(k)}(x)$ as the prediction probability of class $k$ of a given classifier $f_\theta$. For a given sample $(x, y) \sim \mathcal{D}$, the proposed

training objective is as follows:

$$\mathcal{L}_{\text{MART}} := \mathcal{L}_{\text{BCE}}\big(f_\theta(x+\delta), y\big) + \gamma \cdot \big(1 - f_\theta^{(y)}(x)\big) \cdot \text{KL}\big(f_\theta(x) \parallel f_\theta(x+\delta)\big), \tag{10}$$

where $\mathcal{L}_{\text{BCE}} := \mathcal{L}_{\text{CE}}(f_\theta(x), y) - \log(1 - \max_{k \neq y} f_\theta^{(k)}(x))$, $\delta = \arg\max_{\|\delta'\|_p \leq \epsilon} \mathcal{L}_{\text{CE}}\big(f_\theta(x+\delta'), y\big)$, and $\gamma$ is a hyperparameter. For all experiments, we set the hyperparameter $\gamma = 6$ by following the original paper.

**MART with consistency regularization.** One can also utilize $\mathcal{L}_{\text{MART}}$ as the base AT objective and adapt our regularization scheme for the final objective $\mathcal{L}_{\text{total}}^{\text{MART}}$. For a given data augmentations $T_1, T_2 \sim \mathcal{T}$ and sharpening temperature $\tau$, the final objective is as follows:

$$\mathcal{L}_{\text{total}}^{\text{MART}} = \frac{1}{2}\sum_{i=1}^{2}\Big(\mathcal{L}_{\text{BCE}}\big(f_\theta(T_i(x)+\delta_i), y\big) + \gamma \cdot \big(1 - f_\theta^{(y)}(T_i(x))\big) \cdot \text{KL}\big(f_\theta(T_i(x)) \parallel f_\theta(T_i(x)+\delta_i)\big)\Big)$$
$$+ \lambda \cdot \text{JS}\Big(\hat{f}_\theta\big(T_1(x)+\delta_1; \tau\big) \parallel \hat{f}_\theta\big(T_2(x)+\delta_2; \tau\big)\Big), \tag{11}$$

where $\delta_i = \arg\max_{\|\delta\|_p \leq \epsilon} \mathcal{L}_{\text{CE}}\big(f_\theta\big(T_i(x)+\delta\big), y\big)$, and $\hat{f}_\theta(x; \tau)$ is the temperature scaled classifier.

# B. Algorithm

---

**Algorithm 1** Consistency Regularization for Adversarial Robustness

---

**Require:** Batch of samples $\mathcal{B} = \{(x_n, y_n)\}_{n=1}^N$, model $f_\theta$, data augmentation family $\mathcal{T}$, classification loss $\mathcal{L}$, regularization hyperparamater $\lambda$, and sharpening temperature $\tau$

1: **for all** $n \in \{1, ..., N\}$ **do**
2:     Sample $T_1, T_2 \sim \mathcal{T}$ # sample two augmentation funtions
3:     $(\delta_1, \delta_2) \leftarrow (\arg\max_{\|\delta\|_p \leq \epsilon} \mathcal{L}(T_i(x_n), y_n, \delta; \theta))_{i=1}^2$ # perturb each augmentation
4:     $\mathcal{L}_{\text{adv}}^{(n)} \leftarrow \frac{1}{2}\sum_{i=1}^2 \mathcal{L}(T_i(x_n), y_n, \delta_i; \theta)$ # adversarial training with augmentations
5:     $\mathcal{L}_{\text{con}}^{(n)} \leftarrow \text{JS}\Big(\hat{f}_\theta\big(T_1(x_n)+\delta_1; \tau\big) \parallel \hat{f}_\theta\big(T_2(x_n)+\delta_2; \tau\big)\Big)$ # consistency regularization
6:     $\mathcal{L}_{\text{total}}^{(n)} \leftarrow \mathcal{L}_{\text{adv}}^{(n)} + \lambda \cdot \mathcal{L}_{\text{con}}^{(n)}$
7: **end for**
8: $\mathcal{L}_{\text{total}} \leftarrow \frac{1}{N}\sum_{n=1}^N \mathcal{L}_{\text{total}}^{(n)}$
9: $\theta \leftarrow \text{SGD}(\theta, \mathcal{L}_{\text{total}})$

---

**Algorithm 2** Consistency Regularization with Standard Adversarial Training (Madry et al., 2018)

---

**Require:** Batch of samples $\mathcal{B} = \{(x_n, y_n)\}_{n=1}^N$, model $f_\theta$, data augmentation family $\mathcal{T}$, cross-entropy loss $\mathcal{L}_{\text{CE}}$, regularization hyperparamater $\lambda$, and sharpening temperature $\tau$

1: **for all** $n \in \{1, ..., N\}$ **do**
2:     Sample $T_1, T_2 \sim \mathcal{T}$ # sample two augmentation funtions
3:     $(\delta_1, \delta_2) \leftarrow (\arg\max_{\|\delta\|_p \leq \epsilon} \mathcal{L}_{\text{CE}}(f_\theta\big(T_i(x_n)+\delta\big), y_n))_{i=1}^2$ # perturb each augmentation
4:     $\mathcal{L}_{\text{adv}}^{(n)} \leftarrow \frac{1}{2}\sum_{i=1}^2 \mathcal{L}_{\text{CE}}(f_\theta\big(T_i(x_n)+\delta_i\big), y_n)$ # standard adversarial training with augmentations
5:     $\mathcal{L}_{\text{con}}^{(n)} \leftarrow \text{JS}\Big(\hat{f}_\theta\big(T_1(x_n)+\delta_1; \tau\big) \parallel \hat{f}_\theta\big(T_2(x_n)+\delta_2; \tau\big)\Big)$ # consistency regularization
6:     $\mathcal{L}_{\text{total}}^{(n)} \leftarrow \mathcal{L}_{\text{adv}}^{(n)} + \lambda \cdot \mathcal{L}_{\text{con}}^{(n)}$
7: **end for**
8: $\mathcal{L}_{\text{total}} \leftarrow \frac{1}{N}\sum_{n=1}^N \mathcal{L}_{\text{total}}^{(n)}$
9: $\theta \leftarrow \text{SGD}(\theta, \mathcal{L}_{\text{total}})$

---

## C. Detailed description of experimental setups

**Training details.** We use PreAct-ResNet-18 (He et al., 2016) architecture in all experiments, and additionally use WideResNet-34-10 (Zagoruyko and Komodakis, 2016) for white-box adversarial defense on CIFAR-10. We consider augmentation family CCG $\mathcal{T}_{\text{CCG}}$ as in Section 2.2, which consists of random crop (with 4 pixels zero padding), random horizontal flip (with 50% of probability), Cutout (DeVries and Taylor, 2017) (with half of the input width), color jitter, and grayscale for our method. For regularization parameter $\lambda$, we set to $\lambda = 1.0$ in all cases except for applying on WideResNet-34-10 with TRADES and MART where we use $\lambda = 2.0$. The temperature is fixed to $\tau = 0.5$ in all experiments.

For other training setups, we mainly follow the hyperparameters suggested by the previous studies (Pang et al., 2021; Rice et al., 2020). In detail, we train the network for 200 epochs[2] using stochastic gradient descent with momentum 0.9, and weight decay of 0.0005.[3] The learning rate starts at 0.1 and is dropped by a factor of 10 at 50%, and 75% of the training progress. For the inner maximization for all AT, we set the $\epsilon = 8/255$, step size $2/255$, and 10 number of steps with $l_\infty$ constraint (see the supplementary material for the $l_2$ constraint AT results).

**Resource description.** All experiments are processed with a single GPU (NVIDIA RTX 2080 Ti) and 24 instances from virtual CPU (Intel Xeon Silver 4214 CPU @ 2.20GHz).

**Dataset description.** For the experiments, we use CIFAR-10, CIFAR-100 (Krizhevsky et al., 2009), and Tiny-ImageNet.[4] CIFAR-10 and CIFAR-100 consist of 50,000 training and 10,000 test images with 10 and 100 image classes, respectively. All CIFAR images are 32×32×3 resolution (width, height, and RGB channel, respectively). Tiny-ImageNet contains 100,000 train and 10,000 test images with 200 image classes, and all images are 64×64×3 resolution. For all experiments, we do not assume the existence of a validation dataset.

**Data augmentation description.** We use the augmentations family $\mathcal{T}_{\text{base}}$ for baseline adversarial training methods and jointly use $\mathcal{T}_{\text{base}}$, $\mathcal{T}_{\text{CCG}}$ for our regularization. $\mathcal{T}_{\text{base}}$ includes random crop with padding and horizontal flip, and $\mathcal{T}_{\text{CCG}}$ includes Cutout (DeVries and Taylor, 2017), color jitter, and grayscale in addition to the augmentations in $\mathcal{T}_{\text{base}}$. The detailed description of each augmentation in the family are as follows:

- **Random crop with padding.** Randomly crops the padded image with the same size of input image. We zero pad the input image in all sides by 4 pixels.

- **Horizontal flip.** Flips the image horizontally with 50% of probability.

- **Cutout.** Randomly mask out the square regions of input image, with square width of half of the input width.

- **Color jitter.** Change the brightness, contrast, saturation, and hue of the image. We apply color jitter with 80% of probability. We used the same color jitter strength as Chen et al. (2020).

- **Grayscale.** Convert into a gray image. Randomly apply a gray scale with 20% of probability.

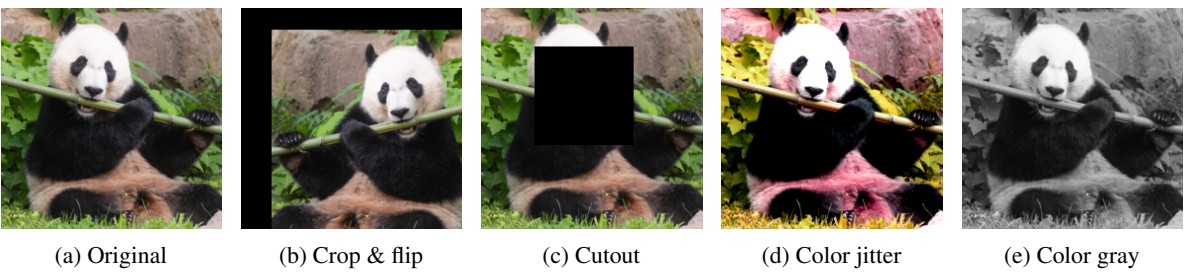

| (a) Original | (b) Crop & flip | (c) Cutout | (d) Color jitter | (e) Color gray |

*Figure 3.* Visualization of a sample image and its data augmentations considered in this paper.

---

[2]Our method maintains almost the same robust accuracy under the same computational budget to the baselines by reducing the training epochs in half. See the supplementary material for more discussion.

[3]On training MART with PreAct-ResNet, we set weight decay to 0.0002 by following the original paper (Wang et al., 2020).

[4]The full dataset of CIFAR, and Tiny-ImageNet can be downloaded at https://www.cs.toronto.edu/~kriz/cifar.html and https://tiny-imagenet.herokuapp.com/, respectively.

*Table 4.* Clean accuracy and robust accuracy (%) against white-box attacks of networks trained on various image classification benchmark datasets. All threat models are $l_\infty$ with $\epsilon = 8/255$. Values in parenthesis denote the result of the checkpoint with the best PGD-10 accuracy, where each checkpoint is saved per epoch. The bold indicates the improved results by our regularization loss.

| Dataset (Architecture) | Method | Clean | PGD-20 | PGD-100 | CW$_\infty$ | AutoAttack |
|---|---|---|---|---|---|---|
| CIFAR-10 (PreAct-ResNet-18) | Standard (Madry et al., 2018) | 84.48 (82.11) | 46.09 (51.75) | 45.89 (51.66) | 45.08 (49.56) | 40.74 (46.43) |
| | + Consistency | **84.65** (83.80) | **54.86** (55.31) | **54.67** (55.08) | **51.32** (51.60) | **47.83** (48.12) |
| | TRADES (Zhang et al., 2019) | 82.20 (82.19) | 51.41 (53.80) | 51.13 (53.59) | 49.04 (51.13) | 46.41 (49.07) |
| | + Consistency | **83.18** (83.18) | **54.86** (54.86) | **54.68** (54.68) | **50.50** (50.50) | **48.30** (48.30) |
| | MART (Wang et al., 2020) | 80.41 (76.95) | 49.60 (52.85) | 49.41 (52.77) | 45.59 (48.46) | 41.89 (46.23) |
| | + Consistency | 80.09 (79.86) | **55.31** (55.30) | **55.16** (55.23) | **50.17** (49.96) | **47.02** (46.71) |
| CIFAR-10 (WideResNet-34-10) | Standard (Madry et al., 2018) | 86.37 (87.55) | 50.16 (55.86) | 49.80 (55.65) | 49.25 (54.45) | 45.62 (51.24) |
| | + Consistency | **88.70** (86.56) | **54.70** (58.28) | **54.53** (58.18) | **54.06** (55.35) | **50.37** (52.16) |
| | TRADES (Zhang et al., 2019) | 85.05 (84.30) | 51.20 (57.34) | 50.89 (57.20) | 50.88 (55.08) | 46.17 (53.02) |
| | + Consistency | **86.39** (86.62) | **57.98** (58.92) | **57.68** (58.58) | **55.96** (56.15) | **52.66** (53.37) |
| | MART (Wang et al., 2020) | 85.75 (83.98) | 49.31 (57.28) | 49.06 (57.22) | 48.05 (53.21) | 44.96 (50.62) |
| | + Consistency | **86.06** (85.91) | **59.90** (61.11) | **59.60** (61.00) | **55.37** (55.84) | **51.46** (52.69) |
| CIFAR-100 (PreAct-ResNet-18) | Standard (Madry et al., 2018) | 56.96 (57.12) | 21.00 (28.98) | 20.86 (28.94) | 21.20 (26.79) | 18.93 (24.24) |
| | + Consistency | **60.21** (60.04) | **28.44** (30.48) | **28.27** (30.43) | **26.44** (27.42) | **23.71** (24.88) |
| Tiny-ImageNet (PreAct-ResNet-18) | Standard (Madry et al., 2018) | 41.10 (44.49) | 10.99 (18.37) | 10.93 (18.34) | 10.79 (16.49) | 9.20 (13.91) |
| | + Consistency | **45.61** (46.50) | **17.76** (20.39) | **17.71** (20.33) | **16.43** (18.01) | **13.56** (15.09) |

# D. More experimental results

## D.1. Main results

**White-box attack.** We consider a wide range of white-box adversarial attacks, in order to extensively measure the robustness of trained models without gradient obfuscation (Athalye et al., 2018): PGD (Madry et al., 2018) with 20 and 100 iterations (step size with $2\epsilon/k$, where $k$ is the iteration number), CW$_\infty$ (Carlini and Wagner, 2017), and AutoAttack (Croce and Hein, 2020b).[5] We report the fully trained model's accuracy and the result of the checkpoint with the best PGD accuracy (of 10 iterations), where each checkpoint is saved per epoch.

As shown in Table 4, incorporating our regularization scheme into existing AT methods consistently improves both best and last white-box accuracies against various adversaries across different models and datasets. The results also demonstrates that our method effectively prevents robust overfitting as the gap between the best and last accuracies has been significantly reduced in all cases. In particular, for TRADES with WideResNet-34-10, our method's robust accuracy gap under the AutoAttack is only 0.71%, while the baseline's gap is 6.85%, which is relatively 10 times smaller. Moreover, we observed that our method almost remains the same test accuracy when even one further train the model, *e.g.*, 50% more training epochs from the current setup.

**Unseen adversaries.** We also evaluate our method against *unforeseen* adversaries including the robustness on different attack radii of $\epsilon$ (the results on different norm constraints of $l_2$ and $l_1$ is already reported in Table 2, nonetheless, we report all values in Table 5 for better presentation).We observe that combining our regularization method could consistently and significantly improve the robustness against all the considered unseen adversaries tested. We remark that there exist fundamental difficulty of achieving the mutual robustness against both $l_1$ and $l_\infty$ attacks (Tramer and Boneh, 2019; Croce and Hein, 2020a). Also, note that it is important to achieve robustness against unseen adversaries which is a more realistic scenario. Hence, we believe our regularization scheme can also be adapted to AT methods for training robust classifiers against multiple perturbations (Tramer and Boneh, 2019; Maini et al., 2020).

**Black-box transfer attack.** We attempt to test the model under black-box transfer attack, *i.e.*, adversarial examples generated from a different model (typically from a larger model). We test PreAct-ResNet-18 trained under baselines and our regularization loss, with crafted adversarial examples from WideResNet34-10 trained with standard AT (we consider PGD-100 and CW$_\infty$ as black-box adversaries). Results in Table 6 demonstrate that our method indeed improves robustness

---

[5]We regard AutoAttack as white-box attack, while it both includes white-box and black-box attacks. We use the standard version from the official implementation: https://github.com/fra31/auto-attack

*Table 5.* Robust accuracy (%) of PreAct-ResNet-18 trained with $l_\infty$ of $\epsilon = 8/255$ constraint against unseen attacks. For unseen attacks, we use PGD-100 under different sized $l_\infty$ balls, and other types of norm ball, *e.g.*, $l_1, l_2$. The bold indicates the improved results by the proposed method.

| Dataset | Method \ $\epsilon$ | $l_\infty$ | | $l_2$ | | $l_1$ | |
|---|---|---|---|---|---|---|---|
| | | 4/255 | 16/255 | 150/255 | 300/255 | 2000/255 | 4000/255 |
| CIFAR-10 | Standard (Madry et al., 2018) | 66.50 | 15.77 | 52.67 | 26.91 | 43.44 | 32.44 |
| | **+ Consistency** | **71.19** | **22.49** | **62.60** | **34.43** | **54.52** | **42.45** |
| | TRADES (Zhang et al., 2019) | 68.47 | 23.87 | 55.91 | 28.31 | 42.36 | 28.64 |
| | **+ Consistency** | **69.82** | **27.18** | **61.46** | **37.11** | **56.09** | **46.73** |
| | MART (Wang et al., 2020) | 66.16 | 20.08 | 55.80 | 30.15 | 43.58 | 27.00 |
| | **+ Consistency** | **67.89** | **27.91** | **60.65** | **38.10** | **54.85** | **43.29** |
| CIFAR-100 | Standard (Madry et al., 2018) | 35.45 | 6.14 | 27.65 | 11.08 | 26.49 | 21.48 |
| | **+ Consistency** | **43.46** | **10.23** | **36.17** | **16.77** | **37.00** | **33.56** |
| Tiny-ImageNet | Standard (Madry et al., 2018) | 22.54 | 2.31 | 27.84 | 17.71 | 32.62 | 30.91 |
| | **+ Consistency** | **29.78** | **4.45** | **34.78** | **28.36** | **38.36** | **36.40** |

*Table 6.* Robust accuracy (%) of PreAct-ResNet-18 against black-box attacks: adversaries are generated from the standard AT (Madry et al., 2018) pre-trained WideResNet-34-10. All models are trained on CIFAR-10. We use $l_\infty$ threat model with $\epsilon = 8/255$. The bold indicates the improved results by the proposed method.

| Method | PGD-100 | CW$_\infty$ |
|---|---|---|
| Standard (Madry et al., 2018) | 69.01 | 79.38 |
| **+ Consistency** | **72.79** | **81.08** |
| TRADES (Zhang et al., 2019) | 69.24 | 77.57 |
| **+ Consistency** | **71.17** | **79.25** |
| MART (Wang et al., 2020) | 68.20 | 75.89 |
| **+ Consistency** | **69.41** | **76.55** |

*Table 7.* Clean accuracy and robust accuracy (%) against diverse attacks of each individual, and combined regularization. The numbers below the attack methods, indicate the radius of the perturbation $\epsilon$. All results are reported on PreAct-ResNet-18 trained under various image classification benchmark datasets. The bold indicates the best results.

| Dataset | Method | Clean | $l_\infty$ (Seen) | | | $l_2$ (Unseen) | | $l_1$ (Unseen) | |
|---|---|---|---|---|---|---|---|---|---|
| | | | PGD-100 (8/255) | CW$_\infty$ (8/255) | AutoAttack (8/255) | PGD-100 (150/255) | PGD-100 (300/255) | PGD-100 (2000/255) | PGD-100 (4000/255) |
| CIFAR-10 | Standard (Madry et al., 2018) | 84.48 | 45.89 | 45.08 | 40.74 | 52.67 | 26.91 | 43.44 | 32.44 |
| | + AWP (Wu et al., 2020) | 79.52 | 53.40 | 50.48 | 47.56 | 59.06 | 32.31 | 47.67 | 32.22 |
| | **+ Consistency** | **84.65** | **54.67** | **51.32** | **47.83** | **62.60** | **34.43** | **54.52** | **42.45** |
| CIFAR-100 | Standard (Madry et al., 2018) | 56.96 | 20.86 | 21.20 | 18.93 | 27.65 | 11.08 | 26.49 | 21.48 |
| | + AWP (Wu et al., 2020) | 52.26 | **29.75** | 26.21 | **24.35** | 35.22 | **20.01** | 34.11 | 31.20 |
| | **+ Consistency** | **60.21** | 28.27 | **26.44** | 23.71 | **36.17** | 16.77 | **37.00** | **33.56** |
| Tiny-ImageNet | Standard (Madry et al., 2018) | 41.10 | 10.93 | 10.79 | 9.20 | 27.84 | 17.71 | 32.62 | 30.91 |
| | + AWP (Wu et al., 2020) | 41.17 | **21.40** | **18.67** | **15.92** | 34.47 | 27.73 | 36.13 | 34.49 |
| | **+ Consistency** | **45.61** | 17.71 | 16.43 | 13.56 | **34.78** | **28.36** | **38.36** | **36.40** |

under black-box attacks across baselines. These results not only imply that our regularization method does not suffer from the gradient obfuscation but also show that our method is effective in practical defense scenarios where the target model is hidden from the attackers.

## D.2. Comparison with Wu et al. (2020)

In this section, we consider a comparison with Adversarial weight perturbation (AWP) (Wu et al., 2020)[6], another recent work which also addresses the overfitting problem of AT by regularizing the flatness of the loss landscape with respect to weights via an adversarial perturbation on both input and weights. We present two experimental scenarios showing that our method can work better than AWP.

---

[6]We use the official implementation: https://github.com/csdongxian/AWP

**White-box attack and unseen adversaries.** We consider various white-box attacks and unseen adversaries for measuring the robustness. As shown in Table 7, our method shows comparable results with AWP in $l_\infty$ defense, and better results in most cases of unseen adversaries defense, *e.g.*, $l_2$, $l_1$ constraint attack. In particular, our regularization technique consistently surpass AWP in the defense against the $l_1$ constraint attack. In addition, our method shows consistent improvement in clean accuracy, while AWP somewhat suffers from the trade-off between clean and robust accuracy.

**Training with limited data.** We also demonstrate that our method is data-efficient: when only a small number of training points are accessible for training the classifier. To this end, we reduce the training dataset's fraction to 10%, 20%, and 50% and train the classifier in each situation. As shown in Figure 4, our method shows better results compare to AWP, especially learning from the small sized dataset, as our method efficiently

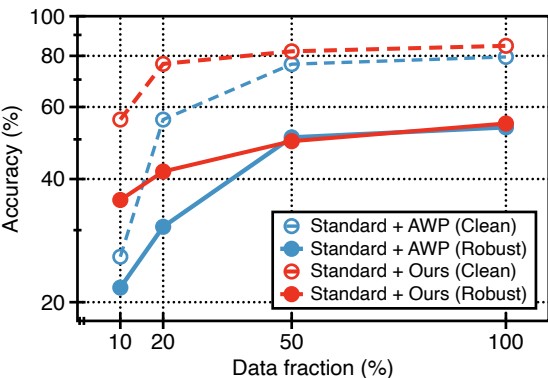

*Figure 4.* Clean accuracy and robust accuracy (%) against PGD-100 attack of $l_\infty$ with $\epsilon = 8/255$, under different fraction (%) of CIFAR-10. We train PreAct-ResNet-18 with AWP (Wu et al., 2020) and consistency regularization loss based on standard AT (Madry et al., 2018).

incorporates the rice space of data augmentations. In particular, our method obtained 35.6% robust accuracy even in the case where only 10% of the total dataset is accessible. We note such efficiency is worthy for practitioners, since in such cases, validation dataset for early stopping is insufficient.

### D.3. Adversarial training on $l_2$ constraint ball

In this subsection, we demonstrate that our regularization scheme is also effective under different types of constraint ball. In particular, we consider adversarial training (AT) on $l_2$ constraint ball.

**Training details.** We use the same training configuration as in Section 3, except for the inner maximization constraint. For inner maximization for all AT, we set the ball radius to $\epsilon = 128/255$, step size $\alpha = 15/255$, and 10 number of steps with $l_2$ constraint. We use PreAct-ResNet-18 (He et al., 2016) architecture in all experiments.

**White-box attack.** For $l_2$ constraint AT, we consider white-box adversarial attacks, including PGD (Madry et al., 2018) with 20 and 100 iterations (step size with $2\epsilon/k$, where $k$ is the iteration number) and AutoAttack (Croce and Hein, 2020b). We report the fully trained model's accuracy and the result of the checkpoint with the best PGD accuracy (of 10 iterations), where each checkpoint is saved per epoch. The results are shown in Table 8. In $l_2$ constraint AT, our regularization scheme also significantly improves white-box accuracy against various adversaries and also reduces the overfitting as the best and last robust accuracy gap has been reduced. Interestingly, our regularization scheme also increases the clean accuracy for $l_2$ constraint AT. This result implies that our method can be a promising method for improving both clean and robust accuracy in various scenarios regardless of the constraint type of the ball.

**Unseen adversaries.** We also evaluate our method against *unforeseen* adversaries, *e.g.*, robustness on different attack radii of $\epsilon$ or even on different norm constraints of $l_\infty$ and $l_1$. As shown in Table 9, our regularization technique also consistently, and significantly improves the robustness accuracy against unseen adversaries, in the case of $l_2$ constraint AT. We believe our method may also improve the robustness against unforeseen attack for other types of constraint ball (*e.g.*, $l_1$). One intriguing direction is to further develop our method toward defending against multiple perturbations (Tramer and Boneh, 2019; Maini et al., 2020) which is also an important research field.

**Common corruption.** We also validate the effectiveness of our method on corrupted CIFAR-10 dataset (Hendrycks and Dietterich, 2019). We report the mean corruption error (mCE) of each model in Table 10. Our method also shows consistent improvement in corruption dataset even for $l_2$ constraint AT. Interestingly, $l_2$ constraint AT shows better performance of mCE compare to $l_\infty$ constraint AT across all corruption types. Moreover, the improvement of our method is more significant in $l_2$ constraint AT. While $l_\infty$ constraint AT's relative improvement is 8.27%, $l_2$ constraint AT shows 14.26% relative improvement in standard AT (Madry et al., 2018). Hence, when the target problem is more focus on the corruption robustness, considering $l_2$ AT can be a one option.

*Table 8.* Clean accuracy and robust accuracy (%) against white-box attacks of networks trained under $l_2$ constraint ball. All threat models are $l_2$ with $\epsilon = 128/255$. Values in parenthesis denote the result of the checkpoint with the best PGD-10 accuracy, where each checkpoint is saved per epoch. The bold indicates the improved results by our regularization loss.

| Dataset (Architecture) | Method | Clean | PGD-20 | PGD-100 | AutoAttack |
|---|---|---|---|---|---|
| CIFAR-10 | Standard (Madry et al., 2018) | 90.17 (89.91) | 63.61 (67.93) | 63.36 (67.77) | 61.88 (65.07) |
| | **+ Consistency** | **91.19** (87.88) | **70.03** (72.77) | **69.85** (72.69) | **68.07** (70.40) |
| | TRADES (Zhang et al., 2019) | 87.19 (87.28) | 65.79 (70.27) | 65.64 (70.14) | 64.28 (68.14) |
| | **+ Consistency** | **88.03** (87.88) | **72.30** (72.77) | **72.23** (72.69) | **70.39** (70.39) |
| | MART (Wang et al., 2020) | 86.36 (86.26) | 64.58 (68.89) | 64.38 (68.75) | 62.63 (65.66) |
| | **+ Consistency** | **87.94** (87.88) | **71.83** (72.70) | **71.73** (72.53) | **68.29** (68.38) |
| CIFAR-100 | Standard (Madry et al., 2018) | 65.94 (66.26) | 36.51 (41.86) | 36.41 (41.64) | 34.98 (37.79) |
| | **+ Consistency** | **67.87** (66.55) | **40.00** (43.33) | **39.85** (43.23) | **37.76** (39.23) |
| Tiny-ImageNet | Standard (Madry et al., 2018) | 55.50 (56.03) | 34.49 (37.19) | 34.38 (37.11) | 33.13 (34.46) |
| | **+ Consistency** | **56.04** (58.84) | **34.95** (39.54) | **34.87** (39.44) | **33.56** (36.99) |

*Table 9.* Robust accuracy (%) of PreAct-ResNet-18 trained with $l_2$ of $\epsilon = 128/255$ constraint against unseen attacks; we use PGD-100 under different sized $l_2$ balls and other types of norm balls, *e.g.*, $l_\infty$, and $l_1$. The bold indicates the improved results by our method.

| Dataset | Method\$\epsilon$ | $l_2$ | | $l_\infty$ | | $l_1$ | |
|---|---|---|---|---|---|---|---|
| | | 64/255 | 256/255 | 4/255 | 16/255 | 2000/255 | 4000/255 |
| CIFAR-10 | Standard (Madry et al., 2018) | 79.06 | 41.98 | 57.67 | 2.96 | 80.60 | 79.05 |
| | **+ Consistency** | **82.80** | 41.25 | **64.76** | **3.59** | **82.08** | **80.92** |
| | TRADES (Zhang et al., 2019) | 78.02 | 41.69 | 61.49 | 10.69 | 78.18 | 77.04 |
| | **+ Consistency** | **81.38** | **51.22** | **67.83** | **12.57** | **81.36** | **80.49** |
| | MART (Wang et al., 2020) | 77.27 | 42.30 | 59.22 | 5.11 | 77.64 | 76.44 |
| | **+ Consistency** | **80.91** | **49.53** | **67.61** | **6.38** | **80.71** | **79.45** |
| CIFAR-100 | Standard (Madry et al., 2018) | 51.34 | 16.65 | 31.00 | 1.61 | 54.32 | 52.82 |
| | **+ Consistency** | **53.83** | **19.64** | **34.54** | **1.83** | **54.33** | **53.00** |
| Tiny-ImageNet | Standard (Madry et al., 2018) | 44.95 | 18.65 | 14.27 | 0.22 | 51.35 | 51.04 |
| | **+ Consistency** | **45.63** | **19.63** | **14.93** | **0.29** | **51.69** | **51.06** |

*Table 10.* Mean corruption error (mCE) (%) of PreAct-ResNet-18 trained on CIFAR-10 under $l_2$ constraint ball, and tested with CIFAR-10-C dataset (Hendrycks and Dietterich, 2019). The bold indicates the improved results by the proposed method.

| Method | mCE ↓ |
|---|---|
| Standard (Madry et al., 2018) | 17.81 |
| **+ Consistency** | **15.27** |
| TRADES (Zhang et al., 2019) | 20.55 |
| **+ Consistency** | **18.21** |
| MART (Wang et al., 2020) | 21.42 |
| **+ Consistency** | **18.38** |

## D.4. Learning dynamics of adversarial training with additional augmentations

Figure 5 shows the test robust and clean accuracy of standard AT (Madry et al., 2018) with additional augmentations from the common practice (*i.e.*, random crop and horizontal flip). We denote such common practice augmentation set as base augmentation. As shown in Figure 5a, further use of Cutout (DeVries and Taylor, 2017) or color augmentation from the base augmentation improves robust accuracy. However, one can observe that it still overfits in the end. In contrast, jointly using two augmentations with the base augmentation can train the classifier without overfitting. We note that the clean accuracy slightly decreases when using CCG augmentation (see Figure 5b), nonetheless, by optimizing our regularization loss simultaneously, the clean accuracy improves from the standard AT with base augmentation (see Table 8).

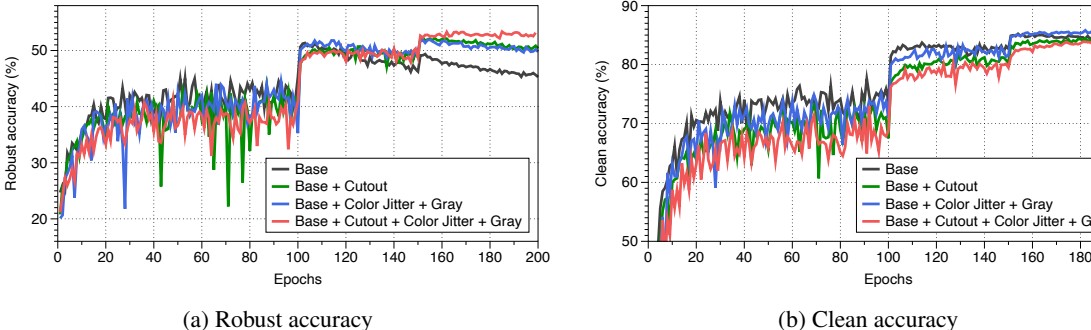

(a) Robust accuracy          (b) Clean accuracy

*Figure 5.* Clean accuracy and robust accuracy (%) against PGD-10 attack of standard AT (Madry et al., 2018) on different augmentations with PreAct-ResNet-18 under CIFAR-10. We use $l_\infty$ threat model with $\epsilon = 8/255$. Base indicates the random crop and horizontal flip.

### D.5. Variance of results over multiple runs

In our experiments, we compare single-run results following other baselines considered in this paper (Madry et al., 2018; Wang et al., 2020). In Table 11, we report the mean and standard deviation of clean and robust accuracy of CIFAR-10 results for standard AT (Madry et al., 2018), and our method. In general, we observe both accuracies of a given training method are fairly robust to network initialization.

*Table 11.* Clean accuracy and robust accuracy (%) against white-box attacks of networks trained under CIFAR-10. All threat models are $l_\infty$ with $\epsilon = 128/255$. The reported values are the mean and standard deviation across 5 seeds. The bold indicates the improved results by our regularization loss.

| Method | Clean | PGD-20 | PGD-100 | CW$_\infty$ | AutoAttack |
|---|---|---|---|---|---|
| Standard (Madry et al., 2018) | $84.44_{\pm0.34}$ | $45.85_{\pm0.25}$ | $45.67_{\pm0.27}$ | $44.91_{\pm0.28}$ | $40.71_{\pm0.28}$ |
| **+ Consistency** | $84.44_{\pm0.23}$ | $\mathbf{54.70}_{\pm0.26}$ | $\mathbf{54.54}_{\pm0.23}$ | $\mathbf{51.52}_{\pm0.23}$ | $\mathbf{47.97}_{\pm0.13}$ |

## E. Ablation study

We perform an ablation study on each of the components in our method. Throughout the section, we apply our method to the standard AT (Madry et al., 2018) and use PGD with 100 iterations for the evaluation.

**Component analysis.** We perform an analysis on each component of our method, namely the use of (a) data augmentations, and (b) the consistency regularization loss, by comparing their robust accuracy and mean corruption error (mCE). The results in Table 12 demonstrate each component is indeed effective, as the performance improves step by step with the addition of the component. We note that the proposed regularization method could not only improve the robust accuracy but also significantly improve the mCE. As shown in Figure 6, simply applying augmentation to the standard AT can reduce the error in many cases (11 types out of 19 corruptions) and even reduce the error of corruptions that are not related to the applied augmentation (*e.g.*, motion blur, zoom blur). More interestingly, further adapting the consistency regularization loss can reduce the corruption error in all cases from the standard AT with augmentation. It suggests that the consistency prior is indeed a desirable property for classifiers to obtain robustness (for both adversarial and corruption).

**Temperature scaling.** We also investigate the effect of the temperature $\tau$ for the consistency regularization. As shown in Table 13, the temperature in our method does matter in the robust accuracy of trained models. As shown in Table 13, sharpening the prediction into more sparse, one-hot like distributions with small temperature $\tau < 1$ on regularization shows an significant improvement. This results somewhat support our claim that the sharp distribution is important in our case, as the confidence (*i.e.*, maximum softmax value) is relatively low on AT than the standard training.

**Analysis on attack directions.** To analyze the effect of our regularization scheme, we observe the attacked directions of the adversarial examples. We find that the most confusing class of the 'clean' input, is highly like to be attacked. Formally, we define the most confusing class of the given sample $(x, y)$ as $\arg\max_{k \neq y} f_\theta^{(k)}(x)$ where $f_\theta^{(k)}$ is the softmax probability of class $k$. We observe that 77.45% out of the misclassified adversarial examples predicts the most confusing class (under standard AT with CCG augmentation). This result implies that the attack direction itself contains the dark knowledge of the given input (Hinton et al., 2015), which supports our intuition to match the attack direction.

*Table 12.* Ablation study on each component of our proposed training objective. Reported values are the robust accuracy (%) against PGD-100 attack of $l_\infty$ with $\epsilon = 8/255$, and mean corruption error (mCE) (%) of PreAct-ResNet-18 under CIFAR-10. The bold indicates the best result.

| Method | PGD-100 | mCE ↓ |
|---|---|---|
| Standard (Madry et al., 2018) | 45.89 | 24.05 |
| + Cutout (DeVries and Taylor, 2017) | 50.51 | 24.11 |
| + CCG Augmentation | 53.22 | 22.87 |
| **+ Consistency** | **54.67** | **22.06** |

*Table 13.* Effect of temperature $\tau$ on robust accuracy (%) against PGD-100 attack of $l_\infty$ with $\epsilon = 8/255$. We train PreAct-ResNet-18 under CIFAR-10 with consistency regularization loss based on standard AT (Madry et al., 2018). The bold indicates the best result.

| $\tau$ | 0.5 | 0.8 | 1.0 | 2.0 | 5.0 |
|---|---|---|---|---|---|
| PGD-100 | **54.67** | 54.21 | 53.82 | 53.47 | 52.82 |

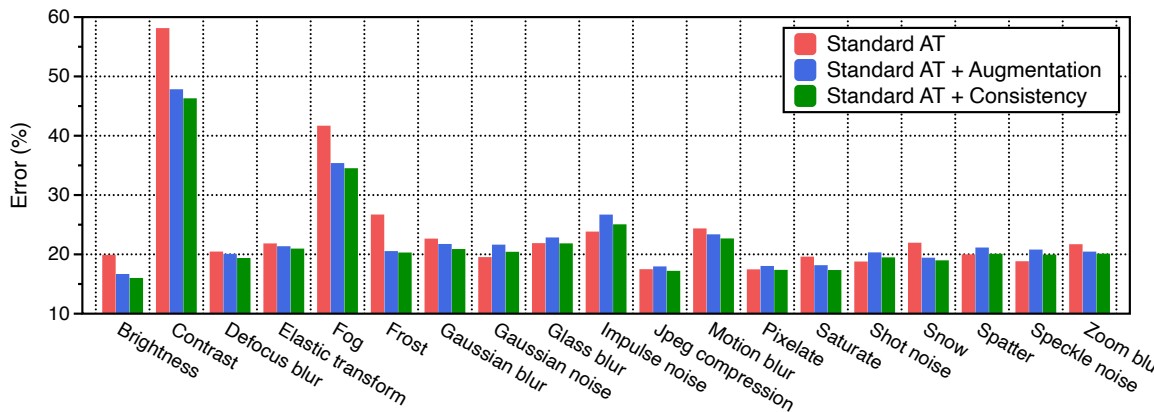

*Figure 6.* Classification error (%) on each corruption type of CIFAR-10-C (Hendrycks and Dietterich, 2019) where the $x$-axis labels denote the corruption type. Reported values are measured on PreAct-ResNet-18 trained under standard AT (Madry et al., 2018), standard AT with proposed augmentation family, standard AT with consistency regularization, respectively.

# F. Additional analysis on the consistency regularization loss

## F.1. Design choices of discrepancy function in the consistency regularization loss

We examine two other popular designs of discrepancy function for the consistency regularization instead of Jensen-Shannon divergence, namely, mean-squared-error and KL-divergence as follow:

$$\mathcal{L}_{\text{MSE}} := \left\| f_\theta\big(T_1(x) + \delta_1\big) - f_\theta\big(T_2(x) + \delta_2\big) \right\|_2^2, \tag{12}$$

$$\mathcal{L}_{\text{KL}} := \text{KL}\Big( f_\theta\big(T_1(x) + \delta_1\big) \parallel f_\theta\big(T_2(x) + \delta_2\big) \Big), \tag{13}$$

where $\delta_i$ is the adversarial noise of $T_i(x)$. We use the same setup as in Section 3 (*e.g.*, use $\mathcal{T}_{\text{CCG}}$) and jointly train the TRADES (Zhang et al., 2019) objective with different choices of consistency losses. The results are presented in Table 14. In general, we observe that the $\mathcal{L}_{\text{MSE}}$ regularizer can improve both clean and robust accuracies, but it could not achieve better robustness than the Jensen-Shannon divergence. Moreover, we observed that $\mathcal{L}_{\text{KL}}$ significantly degrade the robustness against the AutoAttack (Croce and Hein, 2020b). The interesting point is that the $\mathcal{L}_{\text{KL}}$ regularizer extremely lowers the confidence (*i.e.*, maximum softmax probability) of the classifier, *e.g.*, average confidence of the clean examples is 0.39 while other model's confidences are larger than 0.55. Based on our empirical finding, we conjecture that models with very low confidence lead to the venerability against the AutoAttack.

*Table 14.* Comparison with different of discrepancy functions under TRADES (Zhang et al., 2019). Clean accuracy and robust accuracy (%) against white-box attacks of PreAct-ResNet-18 trained on CIFAR-10. We use $l_\infty$ threat model with $\epsilon = 8/255$.

| Discrepancy | Clean | PGD-100 | AutoAttack |
|---|---|---|---|
| None (8) | 82.20 | 51.13 | 46.41 |
| MSE (12) | 83.11 | 54.57 | 48.10 |
| KL-div (13) | 82.70 | 53.79 | 43.39 |
| JS-div (9) | **83.18** | **54.68** | **48.30** |

## F.2. Design choices of data augmentations in the consistency regularization loss

We observed that the design choice of data augmentations in the consistency loss should be done deliberately. To be specific, when utilizing base augmentation (*i.e.*, the augmentation set that cannot prevent overfitting) in the regularization loss, it may induce robust overfitting. To this end, we modify AugMix (Hendrycks et al., 2020), which utilizes base augmentation in the consistency regularization scheme.

We extend AugMix loss to match the attack direction of the given instance. Concretely, for a given sample $(x, y)$, $T \sim \mathcal{T}_{base}$ and $T_1, T_2 \sim \mathcal{T}_{\text{CCG}}$[7], the extension version is:

$$\text{JS}\Big( f_\theta\big(T(x) + \delta\big) \parallel f_\theta\big(T_1(x) + \delta_1\big) \parallel f_\theta\big(T_2(x) + \delta_2\big)\Big), \tag{14}$$

where JS indicates the Jensen-Shannon divergence, $\delta, \delta_1, \delta_2$ is the adversarial noise of $T(x), T_1(x), T_2(x)$, respectively. As shown in Table 15, we find utilizing base augmentation in to the consistency loss, only shows marginal improvement on the adversarial robustness compared to ours. We conjecture that not exposing base augmentations, is crucial for designing the consistency regularization scheme in AT.

*Table 15.* Comparison under different training epoch of standard AT (Madry et al., 2018) with our consistency loss. Last and best robust accuracy (%), against PGD-100 of PreAct-ResNet-18 trained on CIFAR-10. We use $l_\infty$ threat model with $\epsilon = 8/255$.

| Method | Best | Last |
|---|---|---|
| Standard (Madry et al., 2018) | 51.66 | 45.89 |
| + AugMix (14) | 51.77 | 47.08 |
| **+ Consistency** | **55.08** | **54.67** |

## F.3. Runtime analysis

One might concern the training cost of our method, as it is being doubled compared to baseline AT methods due to the two independent adversarial examples. However, we found that our method maintains almost the same robust accuracy even under the same computational budget as the baselines by reducing the training epochs in half. To this end, we train standard AT (Madry et al., 2018) objective jointly with our regularization loss under CIFAR-10. As shown in Table 16, the gap of robust accuracy (between 100 and 200 epoch trained models) under PGD-100, and AutoAttack is only 0.19% and 0.02%, respectively.

*Table 16.* Comparison under different training epoch of standard AT (Madry et al., 2018) with our consistency loss. Robust accuracy (%), against white-box attacks of PreAct-ResNet-18 trained on CIFAR-10. We use $l_\infty$ threat model with $\epsilon = 8/255$.

| Epoch | PGD-100 | AutoAttack |
|---|---|---|
| 100 | 54.48 | 47.81 |
| 200 | 54.67 | 47.83 |

---

[7]AugMix also propose a new augmentation scheme (*i.e.*, mixing augmentation), nonetheless, we only focus on the consistency loss (we observed that mixing augmentation shows similar performance to CCG in AT)