# OpenReview forum: "Consistency Regularization for Adversarial Robustness"
_ICML.cc/2021/Workshop/AML — ICML 2021 Workshop AML Oral_

### Official Review · Reviewer_Ayj8 · 2021-06-20
**The authors propose a consistency regularization loss to address robust overfitting.**

**Rating:** Accept
**Confidence:** 4

**Review:**

- The phenomenon of robust overfitting is a commonly addressed issue in adversarial training (AT). Based on the finding that existing data augmentation (DA) methods are already effective in reducing robust overfitting, the authors propose a new regularization technique for preventing robust overfitting by optimizing an additional consistency loss, which aims to *force the predictor to have similar predictions over adversarial samples from two independent data augmentations of the original input*. And the authors also report robust accuracy improvement on CIFAR-100 and Tiny-ImageNet for both seen and unseen perturbations.
- However, the authors have not provided any theoretical analysis other than empirical results on the constructed regularization term, e.g. convergence proof, so it might be a little intuitive. Also, the legends for Figure 2 are confusing. The name of the figure is "Augmentation removal", while the legends are actually remained augmentations.

---

### Decision · Program_Chairs · 2021-06-21

**Decision:**

Accept (Oral)

**Comment:**

This paper proposed a simple and effective method to address robust overfitting. The proposed method is reasonable. A recent work ("Exploring Memorization in Adversarial Training") also proposed a similar method. It's better to discuss with it.